# Evaluation of Acemannan in Different Commercial Beverages Containing Aloe Vera (*Aloe barbadensis* Miller) Gel

**DOI:** 10.3390/gels9070552

**Published:** 2023-07-06

**Authors:** Francesca Comas-Serra, Paula Estrada, Rafael Minjares-Fuentes, Antoni Femenia

**Affiliations:** 1Department of Chemistry, University of the Balearic Islands, Ctra. Valldemossa km 7.5, C.P. 07122 Palma de Mallorca, Spain; francesca.comas@uib.cat (F.C.-S.); paulaestradagonzalez@gmail.com (P.E.); rafael.minjares@ujed.mx (R.M.-F.); 2Facultad de Ciencias Químicas, Universidad Juárez del Estado de Durango, Av. Artículo 123 s/n, Fracc. Filadelfia, Gómez Palacio 35010, Durango, Mexico

**Keywords:** Aloe vera gel, acemannan, degree of acetylation, cell wall polysaccharides

## Abstract

Aloe vera (*Aloe barbadensis* Miller) gel is a frequently used ingredient in many food pro-ducts, particularly beverages, due to its reported health benefits. Studies have identified acemannan, a polysaccharide rich in mannose units which are partially or fully acetylated, as the primary bioactive compound in Aloe vera gel. The acemannan content and its degree of acetylation (DA) were measured in 15 different commercial beverages containing Aloe vera at varying concentrations (from 30% to 99.8%) as listed on the label. Other biopolymers such as pectins, hemicelluloses, and cellulose were also evaluated. Flavoured beverages (seven samples labelled as containing from 30% to 77% Aloe vera) presented low levels of acemannan (<30 mg/100 g of fresh sample) and were fully deacetylated in most cases. These samples had high levels of other polymers such as pectins, hemicelluloses, and cellulose, likely due to the addition of fruit juices for flavour. Unflavoured beverages (eight samples, with Aloe vera concentrations above 99% according to their labels) had variable levels of acemannan, with only three containing more than 160 mg/100 g of fresh sample. In fact, four samples had less than 35 mg acemannan/100 g of fresh sample. DA levels in all but one sample were lower than 35%, possibly due to processing techniques such as pasteurization causing degradation and deacetylation of the acemannan polymer. Legislation regarding Aloe vera products is limited, and manufacturers are not required to disclose the presence or quality of bioactive compounds in their products, leaving consumers uncertain about the true properties of the products they purchase.

## 1. Introduction

Consumers of all age groups are increasingly choosing natural and organic beverages as they adopt a healthier lifestyle. This shift away from high-calorie carbonated drinks towards plant-based drinks presents an attractive growth opportunity for the market. The demand for plant-based healthy drinks, such as those based on Aloe vera gel, has increased as consumers around the world have become more health conscious. This increased consumer popularity is reflected in the shift in market availability of Aloe beverages from specialty outlets to mainstream groceries and drug stores [1].

The global market for Aloe vera beverages was worth USD 77.8 million in 2019 and an annual growth rate of 11.3% from 2020 to 2027 is projected [2].

Aloe vera (*Aloe barbadensis* Miller) is a popular medicinal plant due to its various health benefits, such as treating burns, reducing fever, improving liver functionality, healing wounds, demostrating antidiabetic and antiviral properties, preventing arthritis, decreasing cholesterol levels, preventing or treating various tumours, and promoting the immune system, among others [3,4,5,6,7,8,9]. Some of these benefits are controversial, with some sources pointing out that the putative effects of Aloe vera are unsupported by clinical studies [10]; however, because consumers are increasingly consuming Aloe vera gel-based beverages, it is important to evaluate the overall quality of these commercial products [11].

Additionally, Aloe vera is a water-dense plant, and Aloe vera gel-based drinks are useful for hydration. The demand for Aloe vera drinks has also been driven by the need for weight management solutions as the number of overweight and obese people has increased globally, leading to a rise in chronic diseases. The COVID-19 pandemic has further increased the demand for plant-based immune-boosting food and beverages, including Aloe vera drinks, as they are a rich source of bioactive components that help enhance immunity by fighting free radical damage in the body [12,13,14]. These trends are also creating new market growth opportunities for Aloe vera gel-based beverages [15].

Drinks containing Aloe vera gel can be classified into two primary categories. One comprises flavoured drinks that generally contain 30–80% Aloe vera gel along with other ingredients such as fruit juices. The other includes unflavoured drinks which contain more than 99% processed Aloe vera gel. These unflavoured Aloe vera drinks emerged as the largest product segment with a share of more than 60% in 2019 and are expected to maintain the lead in the present decade. Moreover, unflavoured drinks are generally consumed as health drinks, maintaining the body well hydrated, providing nutrients, and helping to boost immunity. On the other hand, flavoured Aloe vera gel drinks are anticipated to be the fastest-growing product segment with an annual growth rate of 11.6% from 2020 to 2027.

The Aloe vera plant stores water and various plant nutrients within a clear mucilaginous gel, which is extracted from the parenchymatous cells found in the central area of the leaf cross section [16,17]. The juice intended for oral consumption is primarily obtained from the inner leaf fillet gel, which is also referred to as Aloe vera gel or fillet gel [18]. To obtain this gel, the outer rind and latex are removed, or in the case of whole-leaf juice, the latex material is filtered out using activated charcoal [19]. This purification process is necessary because the latex, which is a separate liquid located between the outer rind and inner fillet gel, contains bitter phenolic molecules such as anthraquinone C- and O- glycosides, anthrones, and some free anthraquinones. The major C-glycoside, aloin A, is the major anthraquinone in Aloe and when oxidized, it produces Aloe-emodin, a free anthraquinone [20].

The Aloe vera gel, as one of the essential components of the Aloe vera plant, comprises primarily water (>98%), with over 60% of the remaining solid being made up of polysaccharides [18]. These polymers can be differentiated into two main groups; on the one hand, a storage polysaccharide known as acemannan, located in the protoplast of the parenchymatous cells of the Aloe vera leaves, and on the other hand, different cell wall polymers such as pectins, cellulose, and hemicelluloses [18,21,22,23,24].

Acemannan, mainly composed of partially acetylated mannose (Man, >60%), glucose (Glc, ~20%), and galactose (Gal, <10%) is regarded as the key functional constituent of Aloe vera [18,25]. The acemannan polysaccharide, with an average molecular weight of 40–60 kDa, has a structural composition that can be represented by a single chain of β-(1 → 4) mannose with β-(1 → 4) glucose inserted into the backbone. In addition, α-(1 → 6) galactose units may be present as side chains [23]. The acetyl groups are the only non-sugar functional groups present in acemannan and seem to play a key role in both the physicochemical properties and the biological activity of Aloe vera gel [22,25,26,27,28].

Therefore, the beneficial properties of food products such as beverages containing Aloe vera gel are attributed to the bioactive compounds derived from the plant, with acemannan being the most important. Thus, the bioactivity of these products may be determined not only by the amount of acemannan but also by its degree of acetylation, which can be seen as a measure of its quality [29,30].

In this context, the main aim of this study was to evaluate the presence of acemannan in a range of commercially available Aloe vera gel-based flavoured and unflavoured beverages from different manufacturers, processors, and food distributors who are prominent in the Aloe vera market at local, national, and international levels. To accomplish this general objective, the following specific objectives were defined: (1) to measure the quantity of acemannan in each analysed beverage, and (2) to assess the bioactivity of acemannan in each sample by determining its degree of acetylation.

## 2. Results and Discussion

### 2.1. Water Content and Dry Residue of Aloe Vera Gel-Based Beverages

The water content or moisture of the different samples was determined after the lyophilization procedure. The results presented in Table 1 show that the water content of the samples ranged from 86.2% (sample 7) to 99.7% (sample 9), with most samples having a moisture percentage of around 98–99%.

Flavoured samples, containing between 30% and 77% Aloe vera gel (samples 1–7), showed greater variation, likely due to the presence of other components such as fruit juices. On the other hand, unflavoured samples, with Aloe vera gel percentages above 99% (as labelled), showed high moisture values (>98.5%) consistent with reported water content values for Aloe vera gel by various authors [13,18].

### 2.2. Alcohol Insoluble Residues (AIRs)

The dry residues obtained from the lyophilization process of the different samples were used to obtain the AIR for each sample.

The extraction yields of the AIRs of all the analysed samples are shown in Figure 1. The yields of the AIRs exhibited great variation, ranging from 15 up to 380 mg AIR/100 g of fresh sample.

Notably, samples 8 and 9 exhibited low AIR content, whereas samples 10 and 13 belonging to the unflavoured samples (labelled as containing more than 99.5% Aloe vera) were the only ones with AIR contents exceeding 300 mg/100 g of fresh sample.

### 2.3. Carbohydrate Composition

The analysed AIRs were found to contain the different types of polysaccharides present in Aloe vera gel, which included the reserve polysaccharide acemannan, as well as characteristic polymers of the cell walls, such as cellulose, hemicelluloses, and pectins [13]. Table 2 shows the individual content of each of the monosaccharides which form those polymers, expressed as mg of sugar/100 g of fresh sample.

In general, the most abundant monomers identified were mannose and/or glucose. The high levels of mannose were likely due to the presence of acemannan, the primary reserve polysaccharide in Aloe vera gel, which confers most of the bioactive properties attributed to the plant [22,31] The high levels of glucose suggest the presence of cellulose, but in some samples, it could also be due to maltodextrin adulteration in the beverage formulation, as observed in the study carried out by Bozzi et al. [32]

The occurrence of small amounts of uronic acids, rhamnose, arabinose, and galactose indicates the presence of pectic polysaccharides, while the presence of other monosaccharides such as xylose and fucose suggests the existence of small amounts of hemicelluloses [18].

### 2.4. Acemannan: Amount and Carbohydrate Composition

As can be observed in Figure 2, all samples containing more than 25 mg of acemannan/100 g of fresh sample were unflavoured Aloe vera gel-based beverages, although samples 8, 9, 12, and 15, exhibited acemannan contents below 30 mg/100 g of fresh sample. On the other hand, within the same group, samples that stood out for their high acemannan content were samples 13, 10, and 11, in that order, with contents exceeding 160 mg/100 g of fresh sample. Here, sample 13 was noteworthy with a content higher than 250 mg/100 g of fresh sample.

Additionally, the flavoured Aloe vera samples in group A presented the lowest acemannan amounts (Figure 2).

One of the main discrepancies found in the different studies involving acemannan, as the main bioactive compound present in Aloe vera gel, lies in its exact chemical composition. In particular, the main differences observed in the scientific literature occur in the percentage of the different monomeric units that form this polysaccharide. Although all the studies agree in highlighting the presence of mannose as the main sugar, the Glc:Man ratio varies from 1:6 to 1:22, showing a predominant presence of mannose accounting for 85 to 96% of the total monomers forming this polysaccharide [21,22,26,27,33,34,35].

The results of the carbohydrate analysis of isolated acemannan from all samples are summarized in Table 3.

As can be observed, mannose was the predominant sugar, followed by glucose and, to a minor extent, galactose. Thus, mannose accounted for 75% to 83% of the total sugars, depending on the sample analysed. Glucose accounted for 16% to 24%, whereas Gal was present in smaller percentages (around 1–2%). These values are similar to those reported by Liu et al. [16], Minjares-Fuentes et al. [22], and Rodríguez-González et al. [36].

Overall, there were no significant differences in the carbohydrate composition of the polymer isolated from the different beverages, either flavoured or unflavoured.

### 2.5. Degree of Acetylation (DA) of Acemannan

Recent studies show that the bioactivity attributed to acemannan depends to a great extent on the DA that this polymer presents. In fact, it has been shown how the deacetylation of this polysaccharide is reflected in the loss or reduction of the bioactivity of this compound [25,26]. Therefore, it is not only important that acemannan is present in significant amounts in commercial Aloe vera gel-based products, but also that this polymer is of the right quality, which is mainly determined by its DA. In fact, all the studies have shown how the functionality of acemannan increases as its DA increases [25,26,28]. Taking this fact into account, the DA of acemannan was determined by ^1^H-NMR in the different commercial beverages analysed.

As can be seen in Figure 3, most of the flavoured samples presented the acemannan polymer fully deacetylated (DA = 0%), with the exception of samples 1 and 7. On the contrary, unflavoured samples exhibited DA of acemannan higher than 10%, such as sample 13, with a DA higher than 99%. This sample not only presented the highest acemannan content compared to the other samples in the same group but also a fully acetylated polymer, which is indicative of its high potential bioactivity.

As an example, Figure 4 shows the ^1^H-NMR spectra of samples of Aloe vera gel-based beverages containing acemannan with DA > 99% (sample 13), DA = 31% (sample 11), and DA = 0% (sample 6), respectively.

Overall, Figure 3 illustrates the notably low quality and the potential lack of bioactivity of the flavoured samples, while unflavoured beverages exhibited relatively higher quality and better potential bioactivity. However, except for sample 13, DA values in the unflavoured group were all below 35%, suggesting deacetylation of acemannan, possibly due to processing conditions of the Aloe vera gel (i.e., use of high pasteurization temperatures).

### 2.6. Cell Wall Polysaccharides Present in the Aloe Vera Gel-Based Beverages

In addition to the acemannan polymer, significant amounts of other polysaccharides were identified in some of the analysed samples. These polymers may have originated not only from the cell walls of the Aloe vera gel but also from the cell walls of other components present in the beverages [37].

In general, from the composition of the sugars present in the AIRs, the presence of cellulose, hemicelluloses, and pectins could be identified (Figure 5).

#### 2.6.1. Cellulose

Cellulose is a structural polysaccharide made up of glucose monomers linked by β-(1,4) glycosidic bonds [38,39]. The content of this polysaccharide in the different analysed samples (expressed as mg of cellulose per 100 g of fresh sample) is shown in Figure 5a.

In general, except for sample 10, the highest presence of cellulose was detected in samples belonging to the group of flavoured beverages. In these samples it is very likely that the considerable amounts of cellulose detected were due to added ingredients such as grape juice in sample 3, or apple juice in the case of sample 6.

#### 2.6.2. Hemicelluloses

Hemicelluloses are a complex group of polysaccharides which contain units of xylose, glucose, and to a lesser extent fucose and mannose [40].

The hemicellulose content of each of the beverage samples is also presented in Figure 5b. As can be observed, the hemicellulose content was highly variable depending on the sample analysed. However, in general, all the samples were characterized by a low presence of fucose, xylose, and galactose, which is indicative of a low presence of xyloglucans, a specific type of hemicellulose common in the cell walls of Aloe vera gel [41,42].

#### 2.6.3. Pectic Polysaccharides

Pectic polysaccharides, or pectins, contain mainly galacturonic acid units in the backbone of their molecular structure, and to a lesser extent, rhamnose. When this polymer is branched, arabinose and galactose units appear [43]. Figure 5c shows the pectin content for each of the analysed samples.

In general, most of the samples presented relatively low amounts of pectins. In the case of sample 6, with a content close to 100 mg/100 g of fresh sample, the presence of a considerable amount of pectins may be due to the addition of apple juice, as it was reflected on its label.

Degree of methylesterification (DME)

Finally, the DME of pectins in each of the samples was also determined by infrared spectroscopy (ATR-FTIR).

Most of the pectins present in the analysed samples exhibited DME values lower than 50% (Figure 6), which indicates that these were low methylation pectins (LMP), except for sample 15, in which pectins exhibited a DME of 52% and therefore could be considered as high methylation pectins (HMP) [44,45].

## 3. Conclusions

From this study conducted with commercial beverages, which, according to the labelling, contain different percentages of Aloe vera gel in their composition (ranging from 30% to 99.8%), the following conclusions can be drawn:

Flavoured samples (seven samples) with a labelled content from 30% to 78% Aloe vera showed very low levels of acemannan (<20 mg acemannan/100 g of fresh sample). Furthermore, in the majority of these seven samples, the detected acemannan was practically deacetylated, as shown by the analyses performed by ^1^H-NMR.

All the flavoured beverages exhibited the presence of relatively significant amounts of other biopolymers such as pectins, hemicelluloses, and cellulose, although this is probably due to the addition of other ingredients in their composition, such as fruit juices.

On the other hand, the acemannan content in the unflavoured samples that, according to the labelling, contained percentages of Aloe vera equal to or higher than 99.5% (eight samples) was highly variable, ranging from 10 to 260 mg acemannan/100 g of fresh sample.

Compared to the previous samples, only three presented acemannan contents higher than 160 mg acemannan/100 g of fresh sample; in fact, in four of them, the determined acemannan content was lower than 35 mg acemannan/100 g of fresh sample.

Additionally, in these eight samples, except for one sample with an acemannan degree of acetylation (DA) higher than 99%, the others showed DA values lower than 35%, which is indicative of their low quality.

Moreover, considering all the examined samples, as can be observed in Figure 7, it is evident that there was no correlation between the acemannan content and the selling price of each individual sample.

In conclusion, due to the lack of current specific legislation that regulates not only the presence but also the quality of bioactive compounds, particularly acemannan in the case of commercial Aloe vera beverages, consumers find themselves in a state of total uncertainty regarding the possible real benefits of these type of products.

## 4. Materials and Methods

### 4.1. Commercial Aloe Vera Gel-Based Beverages

The samples analysed in this study were commercial beverages which contained Aloe vera gel in different percentages (Table 4). All these products were obtained from leading national and international suppliers.

A total of 15 samples were assessed based on the Aloe vera content as indicated on their labels, ranging from 30% to 99.8%. These samples could be categorized into two main groups: the first group (flavoured) consisted of 5 samples with 30% Aloe vera gel (samples 1 to 5) plus two samples with Aloe vera gel contents of 60% and 77% (samples 6 and 7), and the second group (unflavoured) comprised samples 8 to 15, which specified Aloe vera gel contents equal to or greater than 99.5%.

### 4.2. Lyophilization: Dry Residues

All the Aloe vera gel-based samples were lyophilized using a freeze dryer LyoQuest (Telstar, Barcelona, Spain) operated at 0.010 mBar with condenser and shelf temperatures of −80 °C and −20 °C, respectively. In this way, the dry residue of each of the samples was obtained from which all subsequent analyses were carried out. Additionally, the water content present in each sample was estimated.

### 4.3. Alcohol Insoluble Residues (AIRs)

AIRs were obtained by immersing lyophilized juice samples in boiling ethanol (final concentration 85% (*v*/*v*) aqueous) as described by Rodríguez-González et al. [36] with slight modifications.

The lyophilized samples were mixed with 85% *v/v* ethanol and homogenized using an Ultra Turrax T25 Digital (IKA, Staufen, Germany) at 13.000 rpm for 1 min. Then, they were boiled for 5 min to inactivate enzymes present in the samples which could cause degradation of the polysaccharides. The sample was then filtered through a sintered glass filter (Whatman GF-C). The filtrate was again immersed in ethanol (96% *v*/*v*)., homogenized, and extracted with mechanical stirring for 5 min. Next, it was filtered and the solid obtained was extracted again for 5 min with absolute ethanol. Finally, the sample was washed with acetone three times and allowed to dry in the desiccator.

Each AIR was stored in a hermetically sealed container previously identified. The AIRs were used as starting material for the different determinations carried out in the study.

### 4.4. Isolation of Acemannan Polysaccharide

Acemannan isolation was carried out with slight modifications to the method reported by Rodríguez-González et al. [36]. Approximately 150 mg of AIR preparations from Aloe vera gel-based beverages were suspended in 200 mL of distilled water and stirred for 2 h at room temperature. The resulting suspension was subjected to centrifugation at 13,000 *g* for 1 h at 20 °C, yielding supernatants containing acemannan. Next, these supernatants underwent extensive dialysis with a molecular weight cutoff of 10,000–12,000 kDa. Gel permeation chromatography was then employed to further purify the acemannan. Elution of dialyzed fractions containing acemannan was carried out on a Sephacryl S-400-HR column (100 cm × 1 cm) at a flow rate of 16 mL/h. Fractions were dissolved in 50 mM potassium–phosphate buffer (pH 6.5) containing 0.2 M NaCl, and their carbohydrate content was determined using the phenol-sulfuric acid method. Appropriate fractions containing purified acemannan were combined, dialyzed, concentrated, and an aliquot was lyophilized for carbohydrate molecular weight determination and 1H NMR analysis.

### 4.5. Analysis of Carbohydrate Composition

Carbohydrate analysis was performed according to Minjares-Fuentes et al. [30] for neutral sugars after acid hydrolysis with some modifications.

Samples (either AIRs or isolated acemannan) were dispersed in 1 M H_2_SO_4_ and hydrolysed at 100 °C for 2.5 h. Neutral sugars were derivatized and separated in the form of alditol acetates by gas chromatography (GC-FID) following a 21-min temperature program divided into two temperature ramps: the first started at 200 °C, reaching 220 °C after increasing 40 °C/min, then kept for 15 min at 220 °C, and the second ramp began at 220 °C, reaching 230 °C with an increase of 20 °C/min, and was maintained at 230 °C for 5 min. The gas chromatograph (Hewlett-Packard 5890A, Waldbronn, Germany) was equipped with a capillary column DB-225 column (Agilent J&W Scientific, Santa Clara, CA 95051, USA) 30 m long, 0.25 mm internal diameter, and 0.15 μm thick. Helium was used as the carrier gas with a flow rate of 0.8 L/min. The injector temperature was 220 °C while the detector (a flame ionization detector or FID) temperature was 230 °C. The injection volume was set at 1 μL. All determinations were carried out at least in duplicate.

Uronics acids were colorimetrically determined as total uronic acid using a sample hydrolysed for 1 h at 100 °C in 1 M H_2_SO_4_ as described by Blumenkrantz and Asboe-Hansen [46]. Results were expressed as mg of sugar per 100 g of initial dry matter.

### 4.6. ^1^H Nuclear Magnetic Resonance (NMR) Analysis

^1^H NMR analysis of purified acemannan was carried out according to the method proposed by Bozzi et al. [32] with slight modifications reported by Minjares-Fuentes et al. [30] using a Bruker Avance 300 spectrometer (Billerica, MA, USA) equipped with a 5 mm broadband multinuclear z-gradient (BBO) probehead.

Nicotinamide was used to calculate the area under the curve of the corresponding signal of acetyl groups in order to determinate the relative degree of acetylation (DA) of the acemannan polymer. The relative DA of processed samples in relation to the reference sample was calculated using the following Equation (1):
(1)$$\mathrm{Relative}\;\mathrm{degree}\;\mathrm{of}\;\mathrm{acetylation}\;=\left(\frac{A_{processed}}{A_{reference}}\right)\times 100$$
where *A_processed_* and *A_reference_* are the area under the curve of the signals of acetyl groups corresponding to the processed and reference Aloe vera samples, respectively.

### 4.7. Fourier Transformed Infrared (FT-IR) Spectroscopy Analysis

FTIR spectra of the different Aloe vera dehydrated samples (references and juices) were recorded from 4000 to 400 cm^−1^ in an FTIR spectrometer (Bruker Tensor 27, Billerica, MA, USA) at 4 cm^−1^ resolution. Dried samples (~2 mg) were mixed with KBr powder and pressed into pellets for FTIR spectra measurement [47].

### 4.8. Statistical Analysis

The statistical analysis was separately applied to the two groups of analysed samples, namely the group of flavoured samples and the group of unflavoured samples. For the first group (flavoured), letters from ‘t’ to ‘z’ were used to indicate potential significant differences between the samples within that group, while for the second group (unflavoured), letters from ‘a’ to ‘g’ were used for the same purpose.

Data were statistically analysed by one-way analysis of variance (ANOVA). Further, the Tukey–Kramer test was used as a post hoc test with a significant level of *p* < 0.05. All calculations and graphics were carried out using Sigma-plot 10.0 software (Systat Software Inc, Palo Alto, CA, USA).

## Figures and Tables

**Figure 1 gels-09-00552-f001:**
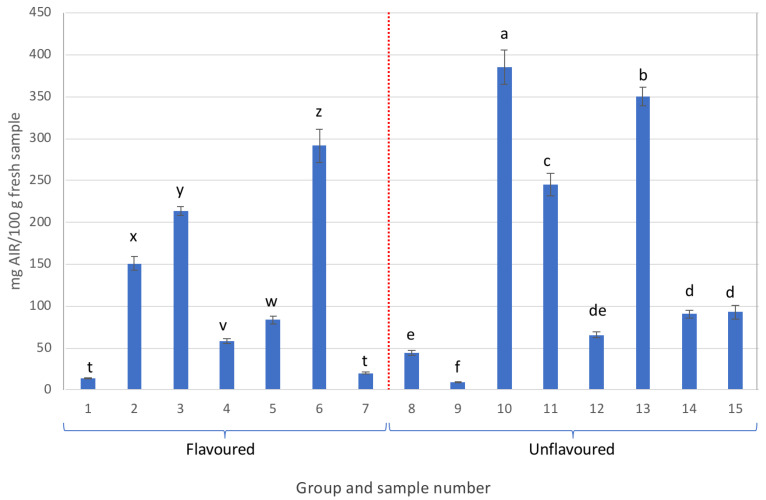
AIR yields obtained from the different samples analysed (different letters in each group, either flavoured or unflavoured, indicates significant differences (*p* < 0.05) between samples of the same group).

**Figure 2 gels-09-00552-f002:**
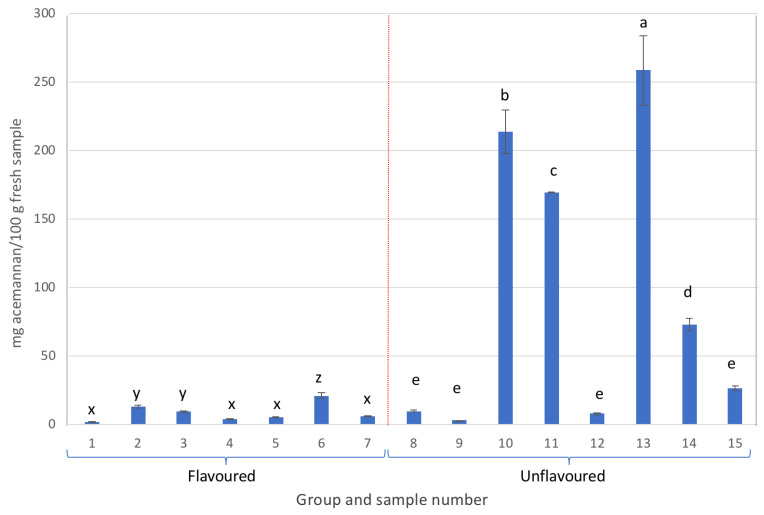
Acemannan content from the different samples analysed (different letters in each group, either flavoured or unflavoured, indicates significant differences (*p* < 0.05) between samples of the same group).

**Figure 3 gels-09-00552-f003:**
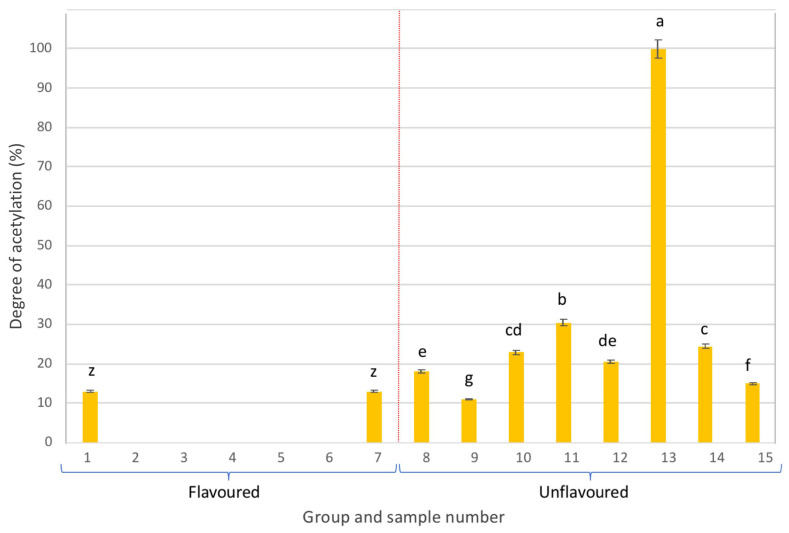
Degree of acetylation of the acemannan polymer present in the different samples analysed (different letters in each group, either flavoured or unflavoured, indicates significant differences (*p* < 0.05) between samples of the same group).

**Figure 4 gels-09-00552-f004:**
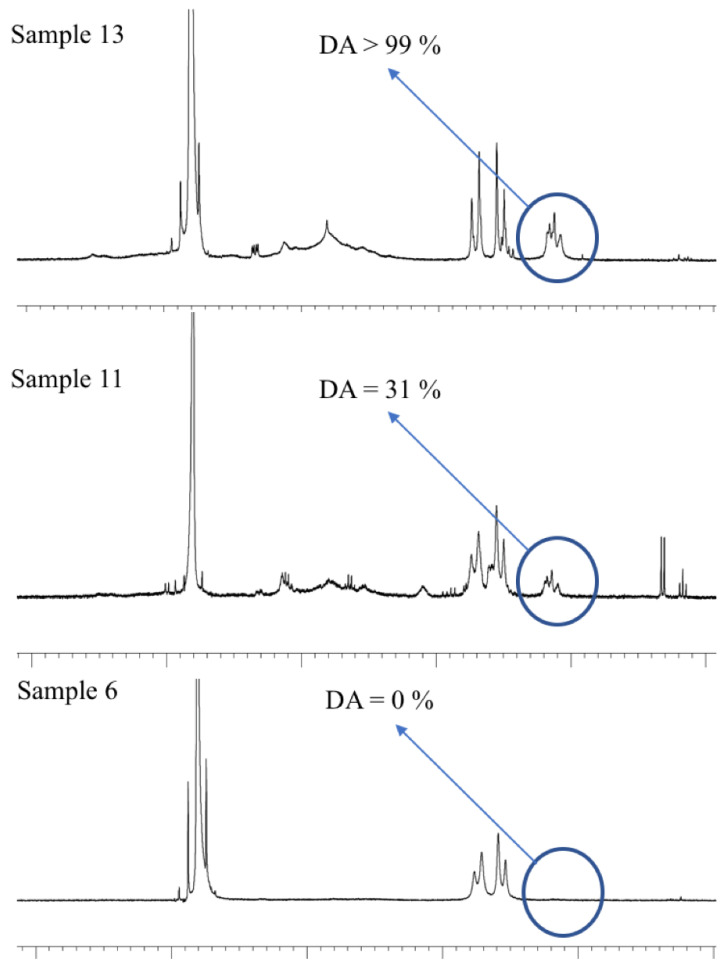
^1^H-NMR spectra of acemannan of Aloe vera gel-based beverage corresponding to samples 13, 11, and 6.

**Figure 5 gels-09-00552-f005:**
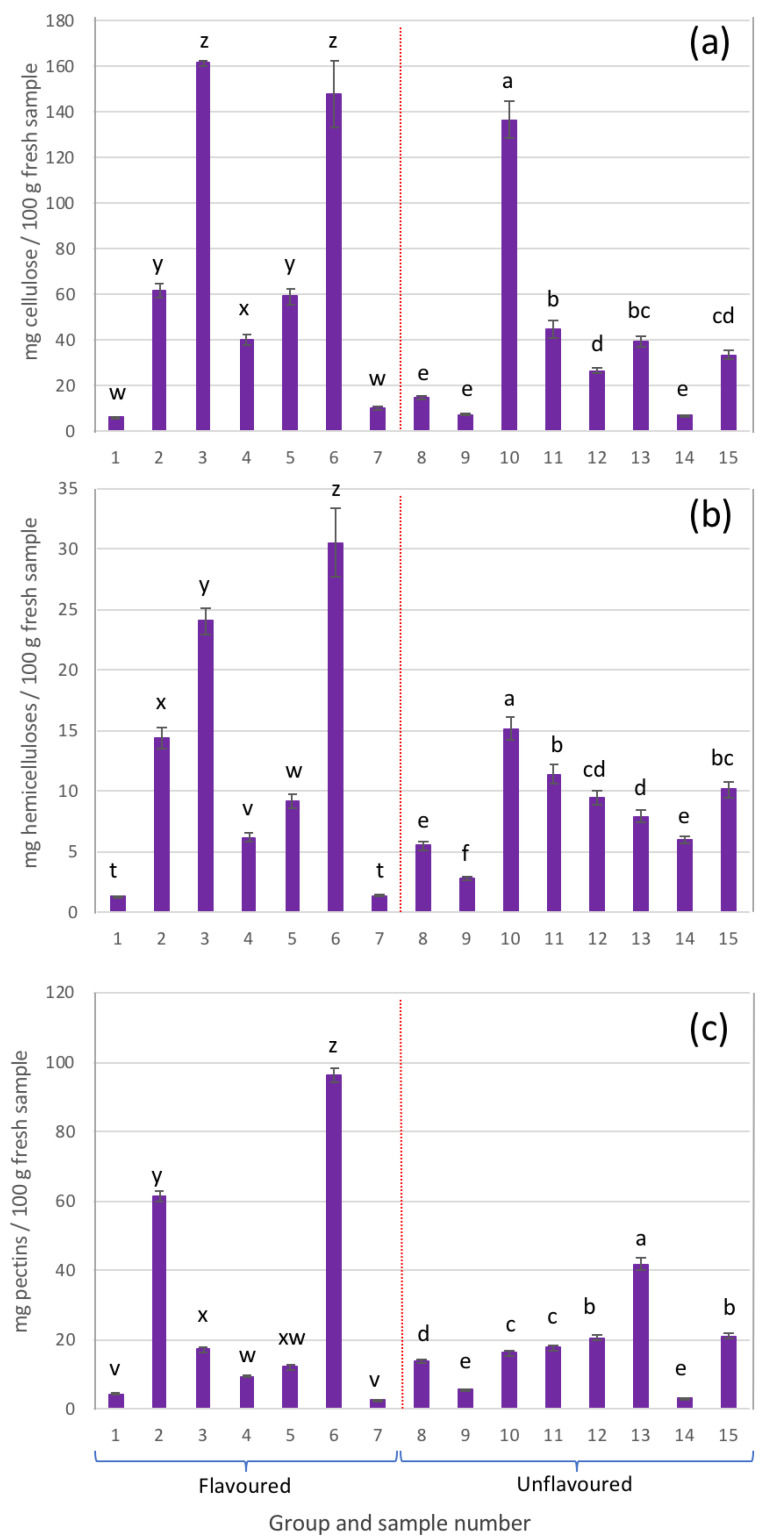
Cell wall polysaccharides: (**a**) cellulose, (**b**) hemicelluloses, and (**c**) pectins present in the different samples analysed (different letters in each group, either flavoured or unflavoured, indicates significant differences (*p* < 0.05) between samples of the same group).

**Figure 6 gels-09-00552-f006:**
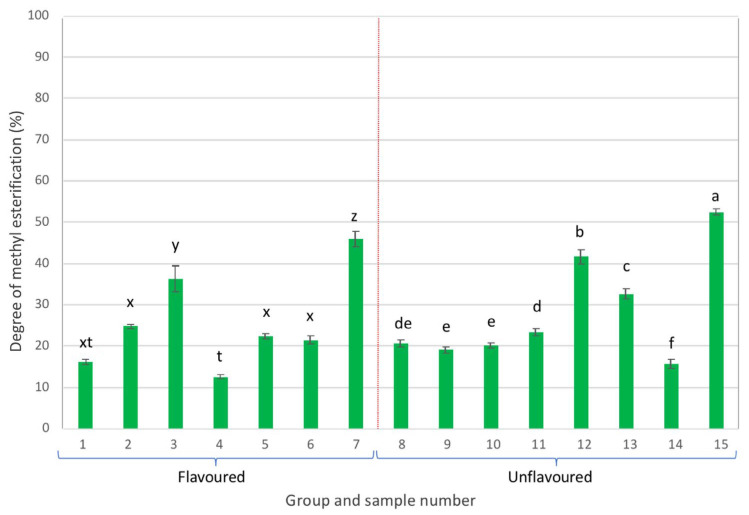
Degree of methylesterification of pectins (different letters in each group, either flavoured or unflavoured, indicates significant differences (*p* < 0.05) between samples of the same group).

**Figure 7 gels-09-00552-f007:**
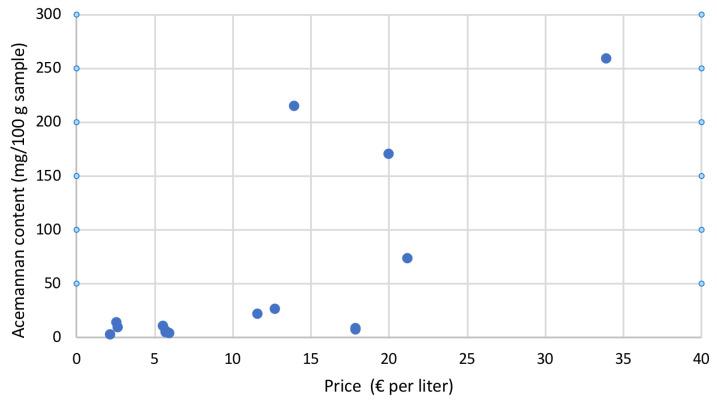
Acemannan content vs. selling price (in euros) per litre of the flavoured and unflavoured commercial Aloe vera gel-based beverages analysed in this study.

**Table 1 gels-09-00552-t001:** Water content (expressed as g H_2_O per 100 g of fresh sample) of the different Aloe vera gel commercial beverages.

Group	Sample Number	Water Content (g H_2_O/100 g Sample)
Flavoured	1	95.92 ± 0.08
	2	99.43 ± 0.05
	3	99.01 ± 0.03
	4	90.34 ± 0.21
	5	89.40 ± 0.14
	6	92.23 ± 0.13
	7	86.21 ± 0.53
Unflavoured	8	99.53 ± 0.05
	9	99.65 ± 0.15
	10	98.64 ± 0.05
	11	99.31 ± 0.07
	12	99.08 ± 0.21
	13	98.92 ± 0.17
	14	99.51 ± 0.04
	15	99.46 ± 0.07

**Table 2 gels-09-00552-t002:** Content of the monosaccharides (mg sugar/100 g of fresh sample) present in the analysed commercial samples of Aloe vera gel-based beverages.

Group	Sample Number	Rha ^1^	Fuc	Ara	Xyl	Man	Gal	Glc	UA
Flav.	1	2.3 ± 0.1	0.1 ± 0.0	0.2 ± 0.0	0.5 ± 0.0	2.0 ± 0.1	0.4 ± 0.0	7.0 ± 0.4	1.5 ± 0.1
	2	4.0 ± 2.7	0.5 ± 0.0	3.3 ± 0.3	7.0 ± 0.6	13.3 ± 0.9	4.0 ± 0.2	68.3 ± 3.3	14.0 ± 1.0
	3	5.8 ± 0.1	0.4 ± 0.0	1.0 ± 0.1	5.7 ± 0.4	9.5 ± 0.6	2.2 ± 0.0	179.2 ± 1.4	8.2 ± 0.6
	4	4.6 ± 0.2	0.3 ± 0.0	0.6 ± 0.0	1.5 ± 0.1	4.0 ± 0.3	0.9 ± 0.0	44.7 ± 2.5	3.4 ± 0.3
	5	4.9 ± 0.2	0.3 ± 0.0	0.6 ± 0.1	2.4 ± 0.1	5.6 ± 0.4	1.2 ± 0.0	65.6 ± 3.7	5.4 ± 0.4
	6	8.1 ± 0.5	1.6 ± 0.1	16.2 ± 1.4	12.5 ± 0.8	21.4 ± 1.8	19.1 ± 0.7	164.2 ± 16.2	52.9 ± 3.1
	7	0.2 ± 0.0	0.0 ± 0.0	0.5 ± 0.0	0.3 ± 0.0	6.5 ± 0.4	0.7 ± 0.0	11.4 ± 0.7	1.4 ± 0.1
Unflav.	8	0.5 ± 0.0	0.4 ± 0.0	1.2 ± 0.1	3.5 ± 0.2	10.1 ± 0.7	2.2 ± 0.1	16.3 ± 0.9	9.8 ± 0.7
	9	0.3 ± 0.0	0.2 ± 0.0	0.4 ± 0.0	1.8 ± 0.1	3.2 ± 0.2	0.8 ± 0.0	8.2 ± 0.5	4.2 ± 0.3
	10	0.0 ± 0.0	0.0 ± 0.0	6.7 ± 0.6	0.0 ± 0.0	214.1 ± 16.0	9.0 ± 0.7	151.8 ± 8.7	0.4 ± 0.0
	11	0.0 ± 0.0	0.0 ± 0.0	5.2 ± 0.4	6.4 ± 0.3	169.7 ± 0.3	6.0 ± 0.2	50.1 ± 4.3	6.6 ± 0.5
	12	0.8 ± 0.0	0.7 ± 0.1	1.1 ± 0.1	5.8 ± 0.4	8.2 ± 0.5	2.2 ± 0.1	29.7 ± 1.7	16.6 ± 1.2
	13	1.5 ± 0.1	0.1 ± 0.0	3.4 ± 0.3	3.5 ± 0.2	258.7 ± 25.6	5.7 ± 0.2	43.9 ± 2.5	31.3 ± 2.9
	14	0.0 ± 0.0	0.0 ± 0.0	0.4 ± 0.0	5.2 ± 0.3	73.1 ± 4.8	1.3 ± 0.1	7.8 ± 0.4	1.6 ± 0.1
	15	1.1 ± 0.0	0.6 ± 0.1	1.9 ± 0.2	5.8 ± 0.4	26.7 ± 1.7	4.6 ± 0.2	37.2 ± 2.1	13.5 ± 1.0

^1^ Rha: rhamnose; Fuc: fucose; Ara: arabinose; Xyl: xylose; Man: mannose; Gal: galactose; Glc: glucose; UA: uronic acids.

**Table 3 gels-09-00552-t003:** Carbohydrate composition (mol%) of acemannan polysaccharide isolated from the Aloe vera gel-based beverages.

Group	Sample Number	Man ^1^	Gal	Glc
Flavoured	1	77.4 ± 1.0	1.1 ± 0.1	21.5 ± 0.5
	2	78.2 ± 0.9	1.0 ± 0.1	20.8 ± 0.8
	3	75.6 ± 1.7	0.7 ± 0.0	23.7 ± 0.6
	4	77.4 ± 0.8	0.6 ± 0.1	22.0± 0.5
	5	78.3 ± 1.0	0.4 ± 0.1	21.3 ± 0.7
	6	77.8 ± 0.9	1.2 ± 0.2	21.0 ± 0.3
	7	77.3 ± 1.8	0.5 ± 0.1	22.2 ± 0.5
Unflavoured	8	79.3 ± 1.7	1.3 ± 0.4	19.4 ± 0.7
	9	79.9 ± 1.2	1.2 ± 0.5	18.9 ± 0.3
	10	77.0 ± 0.9	1.8 ± 0.2	21.2 ± 0.8
	11	78.5 ± 1.3	1.4 ± 0.1	20.1 ± 0.5
	12	76.1 ± 0.9	2.2 ± 0.2	21.7 ± 0.9
	13	82.5 ± 0.7	1.1 ± 0.1	16.4 ± 1.1
	14	79.1 ± 0.9	0.8 ± 0.2	20.1 ± 0.9
	15	78.0 ± 0.7	1.5 ± 0.2	20.5 ± 0.7

^1^ Man: mannose; Gal: galactose; Glc: glucose.

**Table 4 gels-09-00552-t004:** Information on the different commercial Aloe vera gel-based beverages, indicating the group, sample number, % of Aloe vera, price per litre and any additional ingredients present in the beverages.

Group	Sample Number	% Aloe Vera Gel (as Labelled)	Price per Litre (€)	Other Ingredients (as Labelled)
Flavoured	1	30.0	2.18	Fructose, citric acid, sodium citrate, grape aroma, vitamin C, gelan gum, calcium lactate
	2	30.0	2.60	Fructose, citric acid, sodium citrate, grape aroma, vitamin C, gelan gum, calcium lactate
	3	30.0	2.70	Fructose, citric acid, grape juice, ascorbic acid, carboxymethyl cellulose, steviol
	4	30.0	5.70	Fructose, calcium lactate, citric acid, sodium citrate, grape flavour, sucrose, gelan gum
	5	30.0	5.70	Fructose, calcium lactate, citric acid, sodium citrate, pomegranate flavour
	6	60.0	11.63	Apple juice, sucralose, citric acid, potassium sorbate, sodium benzoate
	7	76.8	17.90	Pomegranate juice, citric acid, ascorbic acid
Unflavoured	8	99.5	5.55	Citric acid, ascorbic acid
	9	99.5	5.99	Citric acid, ascorbic acid
	10	99.7	13.99	Citric acid, potassium sorbate, ascorbic acid
	11	99.7	19.99	Citric acid
	12	99.8	17.90	Citric acid, ascorbic acid
	13	99.8	33.90	Citric acid
	14	99.8	21.20	Citric acid
	15	99.8	12.75	Citric acid, green tea, stevia

## Data Availability

Data is contained within the article.

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
