# Peer review of "Evaluation of Acemannan in Different Commercial Beverages Containing Aloe Vera (Aloe barbadensis Miller) Gel"

_gels, 2023, doi:10.3390/gels9070552_

Round 1
Reviewer 1 Report
Aim of this study is ambiguous and need clarification.
Different samples of beverage contain varying amount of aloe vera. What the reasons to carry out this research. Therefore, it can be concluded that composition is dependent on Aloe vera gel's concentration. The novelty of the research is not described.
Its ok
Author Response
Please find our reply to reviewer 1 in the attached document

Reviewer 2 Report
This an interesting article evaluating the acemannan concentration in different commercial beverages containing Aloe vera gel; however, three major issues still need to be addressed: 1] What is the bioactivity of acemannan? and 2] How was the Aloe vera gel prepared for inclusion into the beverage? and 3] How was the Aloe vera gel determined to be isolated from Aloe barbadensis Miller?
ABSTRACT
What is the basis for suggesting that acemannan is the primary active ingredient in Aloe vera gel?
INTRODUCTION
What is the basis for suggesting that acemannan is the primary active ingredient in Aloe vera gel? What about anthraquinones?
What is the bioactivity of acemannan; why do the authors consider it the most bioactive? Please add supportive references.
page 2/16: an storage should be a or the storage
Why are Materials and Methods at the end of the paper?
RESULTS AND DISCUSSION
TABLE 1: What other fruit juices?
line 114: was should be were
Figure 1: (same for figure 2) different letters in each group indicates significant difference between samples; between which samples? [samples 1 & 7, flavored with letter t not different from each other but different from samples 2,3,4,5, & 6?] Sample 6 with a Z is different from all the other flavored samples?] If this is correct, it should be explained in the figure legend; if not, the letters need to be explained in the figure legend
Table 2. Carbohydrate composition is all over the place---any meaning? Is the table need or can it be replace with a sentence or two?
6/16, line 184: In fact, all studies..............................need references.
Figure 3. Does this mean that the unflavored beverages are more "bioactive"?
MATERIALS AND METHODS
How processed? decolored or filet? Purpose of Table 4?
OK
Author Response
Please see answer to reviewer 2 in the attached document

Round 2
Reviewer 1 Report
Authors has improved the article according to the suggestions of the reviewers and therefore accepted.
in Acceptable form
Reviewer 2 Report
Thanks for your responses to the questions raised.